# Reduced midbrain raphe echogenicity in patients with fibromyalgia syndrome

**Nurcan Üçeyler**[1]*, **Mira Schließer**[1], **Dimitar Evdokimov**[1], **Jakub Radziwon**[2¤], **Betty Feulner**[1], **Stefan Unterecker**[3], **Florian Rimmele**[2], **Uwe Walter**[2]

**1** Department of Neurology, University of Würzburg, Würzburg, Germany, **2** Department of Neurology, Rostock University Medical Center, Rostock, Germany, **3** Department of Psychiatry, Psychotherapy and Psychosomatic Medicine, University of Würzburg, Würzburg, Germany

¤ Current address: Student Scientific Group of Neurology, Faculty of Medicine, Medical University of Gdańsk, Gdańsk, Poland
* ueceyler_n@ukw.de

**Citation:** Üçeyler N, Schließer M, Evdokimov D, Radziwon J, Feulner B, Unterecker S, et al. (2022) Reduced midbrain raphe echogenicity in patients with fibromyalgia syndrome. PLoS ONE 17(11): e0277316. https://doi.org/10.1371/journal. pone.0277316

**Data Availability Statement:** All relevant data are within the paper and its Supporting Information files.

## Abstract

### Objectives

The pathogenesis of fibromyalgia syndrome (FMS) is unclear. Transcranial ultrasonography revealed anechoic alteration of midbrain raphe in depression and anxiety disorders, suggesting affection of the central serotonergic system. Here, we assessed midbrain raphe echogenicity in FMS.

### Methods

Sixty-six patients underwent transcranial sonography, of whom 53 were patients with FMS (27 women, 26 men), 13 patients with major depression and physical pain (all women), and 14 healthy controls (11 women, 3 men). Raphe echogenicity was graded visually as normal or hypoechogenic, and quantified by digitized image analysis, each by investigators blinded to the clinical diagnosis.

### Results

Quantitative midbrain raphe echogenicity was lower in patients with FMS compared to healthy controls (p<0.05), but not different from that of patients with depression and accompanying physical pain. Pain and FMS symptom burden did not correlate with midbrain raphe echogenicity as well as the presence and severity of depressive symptoms.

### Conclusion

We found reduced echogenicity of the midbrain raphe area in patients with FMS and in patients with depression and physical pain, independent of the presence or severity of pain, FMS, and depressive symptoms. Further exploration of this sonographic finding is necessary before this objective technique may enter diagnostic algorithms in FMS and depression.

**Funding:** The study was supported by intramural funds of the University of Würzburg and by the Else Kröner-Fresenius-Stiftung (N.Ü.: 2014_A129). N.Ü. was funded by Deutsche Forschungsgemeinschaft (DFG; UE171/15-1). The funders had no role in study design, data collection and analysis, decision to publish, or preparation of the manuscript.

## Introduction

Chronic widespread pain with additional key symptoms such as depressed mood, fatigue, and sleep disturbance characterizes the clinical phenotype of patients with fibromyalgia syndrome (FMS) [1]. Several studies have provided evidence for an involvement of the central serotonergic system in FMS, including alterations of the hypothalamic-adrenergic axis [2] and gene polymorphisms found in serotonergic genes [3–5]. Further, while several studies suggest an impact of the peripheral nervous system on symptom development in FMS [6, 7], there are also studies supporting a crucial role of the central nervous system [8].

Transcranial sonography revealed hypo- or anechoic alteration of the midbrain raphe in the majority of patients with major depressive or panic disorder [9–13]. These findings were associated with a lesion or dysfunction of serotonergic raphe nuclei [14, 15]. More recently, lower midbrain raphe echogenicity was also found in patients with chronic pain conditions such as migraine [16, 17] or burning mouth syndrome [18].

The aim of our study was to investigate the diagnostic utility of midbrain raphe sonography when assessing patients with FMS.

## Patients and methods

### Recruitment of study cohort

From September 2014 to October 2022, we registered 53 patients with FMS (27 women, 26 men; median age 55 years, range 33–75 years) into a long-term study cohort [6] at the Department of Neurology, University of Würzburg, Germany, diagnosed according to current criteria and after exclusion of differential diagnoses [1, 19, 20]. We further included 13 women with major depression and additional physical pain (median age 52 years, range 32–58 years) at the Department of Psychiatry, University of Würzburg, Germany according to the Diagnostic and Statistical Manual of Mental Disorders-IV. Fourteen healthy controls (eleven women, three men; median age 49 years, range 26–64 years) were recruited at the Department of Neurology, University of Würzburg, Germany and served as controls. Our study was approved by the Ethics Committee of the University of Würzburg Medical Faculty (#121/14, #170/18) and all study participants gave written informed consent before inclusion.

### Clinical examination

Patients with FMS underwent a detailed medical interview and complete neurological examination. Patients with depression and physical pain also underwent detailed medical interview, complete neurological examination, and psychiatric assessment. All patients completed the Neuropathic Pain Symptom Inventory (NPSI) [21] and the Graded Chronic Pain Scale (GCPS) [22] for characterization of pain. Also, all patients filled in the "Allgemeine Depressionsskala" (ADS), which is a screening tool for the presence of depressive symptoms [23].

### Transcranial midbrain raphe sonography

Transcranial sonography of midbrain raphe was performed in all study participants by an experienced investigator (M.S.) blinded to subject group allocation and in random order. For this, a Toshiba Aplio XG ultrasound system equipped with a 2.5 MHz probe was employed. Assessments on B-mode imaging were performed in axial plane and bilaterally at the temporal acoustic bone window with a penetration depth of 14 cm and a dynamic range of 50 dB. Representative video clips and images of the midbrain raphe region were recorded bilaterally for each participant to enable off-line image analysis.

### Data assessment

The investigator performing the off-line analysis was unaware of participants'group allocation. The stored ultrasound images and clips were completely reviewed for each participant. Midbrain raphe echogenicity was graded visually on a semi-quantitative scale by an experienced investigator (U.W.) as described earlier in detail [17]: grade 1, no echosignal; grade 2, reduced (low or interrupted) echosignal; grade 3 high (normal) echosignal. To assess interrater reliability of visual semi-quantitative ratings, midbrain raphe echogenicity of each participant was graded independently by a second well-trained reader (J.R.).

To exactly quantify the echo-intensity of midbrain raphe, off-line digitized image analysis was performed using a validated MATLAB-based software tool [13]. This tool first pre-assesses the overall image quality before starting the analysis of distinct structures. Then, a normalized echo-intensity measure of the target structure in the referring region of interest was achieved. For each case, minimum two to four images per side of insonation were analyzed and the mean value of raphe echo-intensity was calculated for each side. The mean value of bilateral measures was then used for further analysis.

### Statistical analysis

For statistical analysis, SPSS Statistics 28 software (IBM, Ehningen, Germany) was used. The non-parametric Mann-Whitney U test was applied when data was not normally distributed. Correlations were calculated using the bivariate Spearman correlation coefficient. Statistical significance was assumed at $p<0.05$. Interrater reliability of visual assessment of midbrain raphe echogenicity as normal or abnormal (reduced) was assessed by Cohen's kappa.

## Results

### Clinical characterization

Table 1 summarizes the main clinical characteristics of the study cohort. Fig 1 shows representative midbrain sonograms for the three study groups. While pain in FMS mostly manifested as generalized, deep-aching pain often described like muscle soreness, patients with depression predominantly reported bilateral, but focal pain primarily affecting the back.

### Midbrain raphe echo-intensity is reduced in patients with FMS compared to healthy controls, and does not differ from patients with depression and physical pain

Quantitatively assessed echo-intensity of midbrain raphe was lower in patients with FMS compared to controls ($p<0.05$; Fig 2). Midbrain raphe echo-intensity was also reduced in patients with depression and physical pain compared to healthy controls ($p<0.05$; Fig 2). No intergroup difference was found between the two patient groups and also not between women (median midbrain raphe echo-intensity: 27.5, range 18.4–34.8) and men with FMS (median midbrain raphe echo-intensity: 25.3, range 15.4–40.5; $p>0.05$). Analysis of visual assessment of midbrain sonograms yielded analogous results with no intergroup difference between the two patient groups (FMS: 3, range 2–3; depression and pain: 3, range 1–3) (Table 1). Also, no difference was found between women and men with FMS (3, range 2–3 each). Interrater reliability of visual assessment of midbrain raphe was high (Cohen's kappa = 0.85, $p<0.001$).

**Table 1. Study subjects.**

|  | Healthy controls | Depression+pain | FMS | p-value |
|---|---|---|---|---|
|  | I | II | III | II vs III |
| N (women) | 14 (11) | 13 (13) | 53 (27) |  |
| Age [range] | 49 [26–64] | 52 [32–58] | 55 [33–75] | n.s. |
| Pain duration | NA | 5 (1–37) | 20 (2–60) | <0.05 |
| NPSI sum score | NA | 30 (0–70) | 30 (10–100) | n.s. |
| GCPS current pain | NA | 6 (3–10) | 5 (0–9) | n.s. |
| ADS score | NA | 41 (22–49) | 23 (6–51) | <0.001 |
| FIQ score | NA | 43 (6–71) | 46 (9–66) | n.s. |
| WPI | NA | NA | 15 (8–19) |  |
| SSS | NA | NA | 8 (3–12) |  |
| Number of patients taking serotonergic medication | NA | 12/13 (92%) | 3/53 (6%) |  |
| Number of patients with migraine | NA | 1/13 (8%) | None |  |
| Raphe echogenicity (qualitative) | 3 (2–3) | 3 (1–3) | 3 (2–3) | n.s. |
| Raphe echo-intensity (quantitative) | 30 (23–44) | 27 (19–34)# | 26 (15–40)## | n.s. |

**Abbreviations:** ADS = Allgemeine Depressionsskala, FIQ = Fibromyalgia Impact Questionnaire, GCPS = Graded Chronic Pain Scale, NPSI = Neuropathic Pain Symptom Inventory, SSS = Symptom Severity Score, WPI = Widespread Pain Index.

#I vs II: p<0.05.

##I vs III: p<0.05.

## Midbrain raphe echo-intensity positively correlates with FMS symptom severity, but not with depressive symptoms

In FMS, midbrain raphe echo-intensity positively correlated with symptom severity as reflected by the SSS (correl. coeff.: 0.344; p<0.05; Fig 3), which was not assessed in patients with depression. In both patient groups, severity of depressive symptoms as assessed via the ADS did not correlate with midbrain raphe echogenicity (Fig 3). The same was true for pain severity and duration (S1 and S2 Tables). Interestingly, FIQ scores positively correlated with the ADS score, which was more prominent in FMS patients (correl. coeff.: 0.586; p<0.01) than in patients with depression and physical pain (correl. coeff.: 0.555; p = 0.049; Fig 3).

## Discussion

We investigated midbrain raphe echogenicity in patients with FMS and compared data with those of patients with depression and physical pain, and with healthy controls. Our data show a reduction in midbrain raphe echogenicity in FMS patients that is similar to findings in patients with depression and physical pain. Interestingly, raphe echogenicity positively correlates with symptom severity in FMS patients as reflected by the SSS and midbrain raphe echogenicity does not correlate with the presence and severity of depressive symptoms.

After introduction of transcranial midbrain raphe ultrasonography in patients with depression [9, 10], the technique was applied in various neurological and psychiatric disorders. The finding of reduced echogenicity in the midbrain raphe region is well reproduced in patients with depression [15, 24–26], migraine [16], panic disorders [13], and obsessive-compulsive disorder [27]. Several studies also reported a correlation of raphe echogenicity with additional depressive symptoms in the respective primary disease such as in epilepsy [28], Parkinson's disease [29] or migraine [30, 31]. However, data are controversial [12].

FMS is a chronic pain syndrome of unknown pathophysiology with contributions from the peripheral and central nervous system [7, 32]. Depressive symptoms are frequently reported

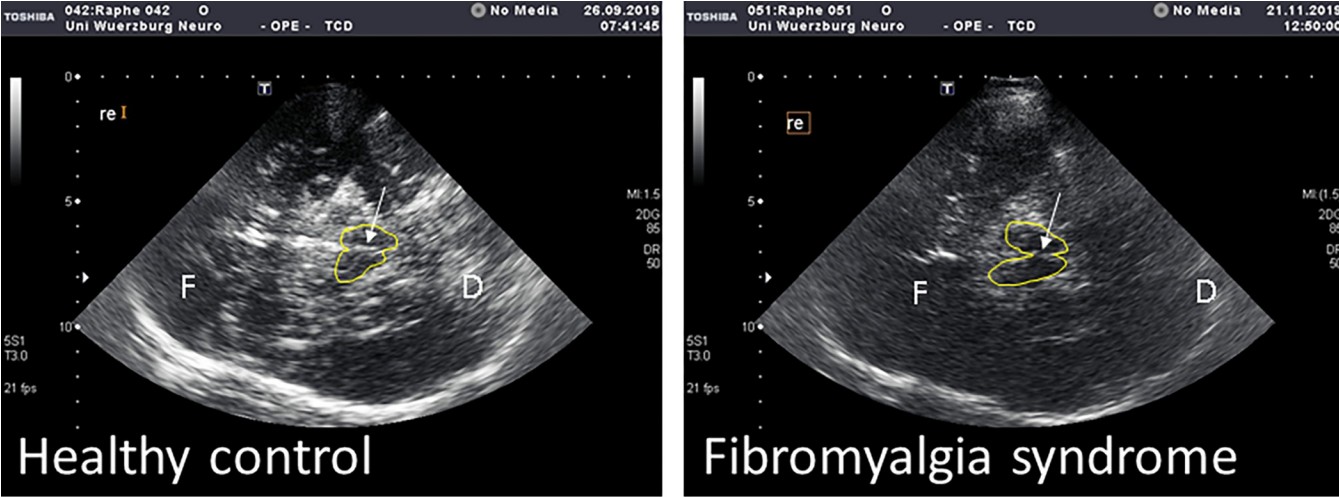

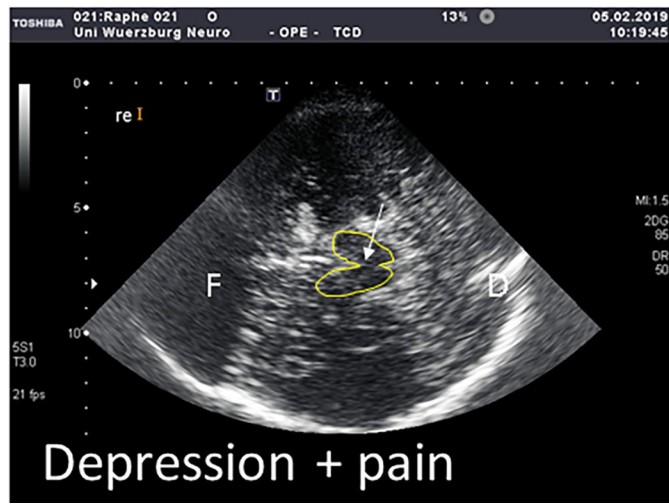

**Fig 1. Exemplary midbrain raphe sonograms.** Transcranial axial sonogram of the midbrain in (A) a healthy woman, (B) a woman with fibromyalgia syndrome, and (C) a woman with depression and physical pain. Midbrain raphe echogenicity (arrow) is markedly lower in B and C compared to A. The midbrain is outlined in yellow. For orientation: F = frontal, D = dorsal.

by patients with FMS, while the diagnostic criteria are distinct [1, 19] and there is ample evidence for a differentiation between depression and FMS with regard to signs and symptoms of the nervous system [6, 33]. When investigating the midbrain raphe region, we found a reduction of echogenicity that was not different from that found in our disease control group of women with depression and additional physical pain. Interestingly, sonography results did not correlate with the presence and severity of depressive symptoms as quantified using the ADS sum score which was higher in patients with depression than in FMS patients (Table 1). Hence, cross-correlations are not possible.

The pathophysiological mechanism underlying alterations in midbrain echogenicity remain unclear. One hypothesis is that a reduction of serotonergic neurons or their impaired functionality may cause hypoechogenicity in this brain region. Also, differences in the response rate to serotonergic medication was found when assessing raphe echogenicity in correlation with patients'medication [15]. In our study, we cannot draw robust conclusions in this context since only three patients with FMS took serotonergic medication. On the contrary, the

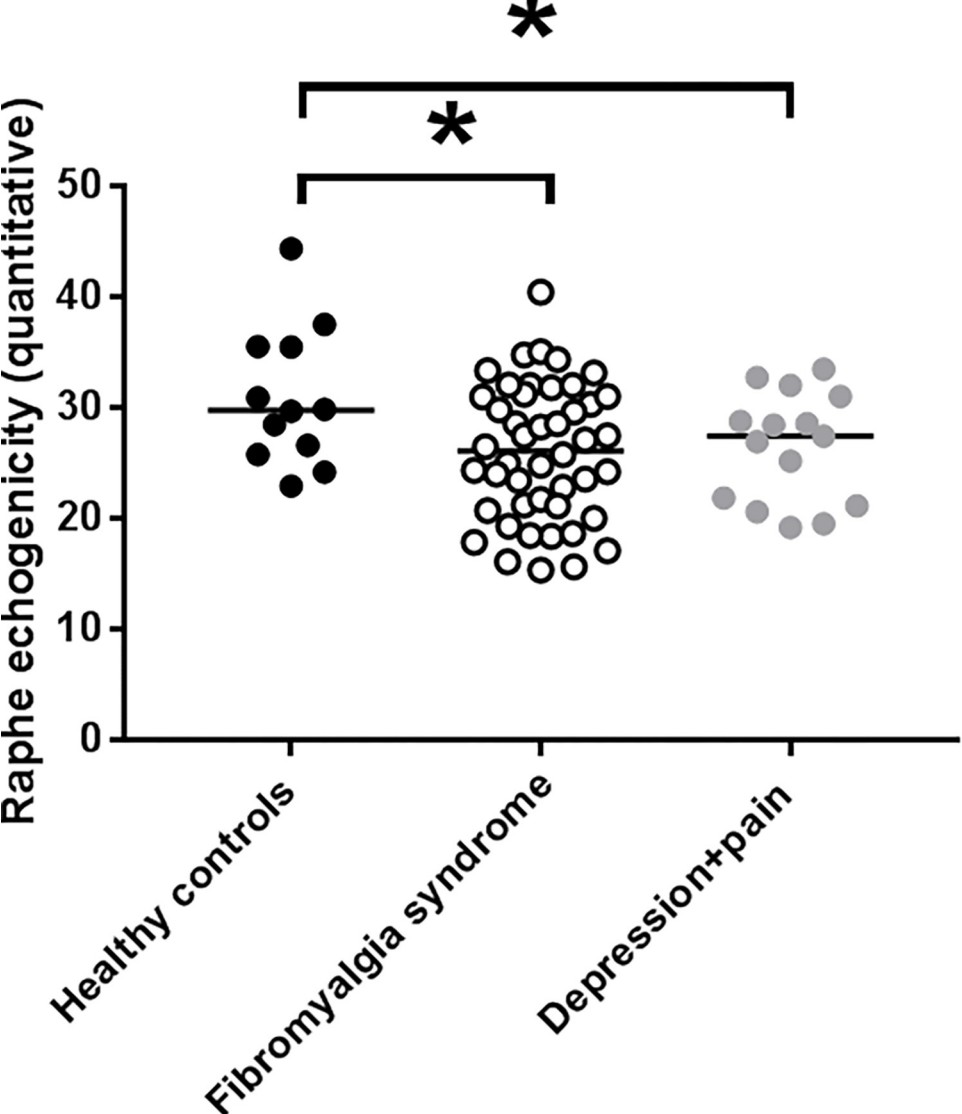

**Fig 2. Quantitative midbrain raphe echo-intensity.** The graph displays the individual mean values of raphe echo-intensity obtained by digitized image analysis in healthy controls, patients with fibromyalgia syndrome, and patients with depression and physical pain. Midbrain raphe echo-intensity was lower in patients with fibromyalgia syndrome compared to healthy controls, and in patients with depression and physical pain compared to healthy controls (each, $p < 0.05$). No intergroup difference was found between the two patient groups.

majority of patients with depression were on selective serotonin reuptake inhibitors, however, in almost all cases this medication was combined with other antipsychotic and/or antidepressant drugs such that the pure effect of serotonin reuptake inhibitors could not be differentiated (Table 1).

While midbrain raphe echogenicity did not correlate with pain characteristics of our patient groups, we found a positive correlation with FMS disease severity as reflected by the SSS [1]. While the SSS was not assessed in patients with depression, this finding is of note and may potentially point towards diversity in the pathomechanism of FMS and depression. As for the seemingly contradictory data in our study to published literature regarding reduced midbrain echogenicity in patients with depressive symptoms it is of note that we investigated patients

## FMS

## Depression+pain

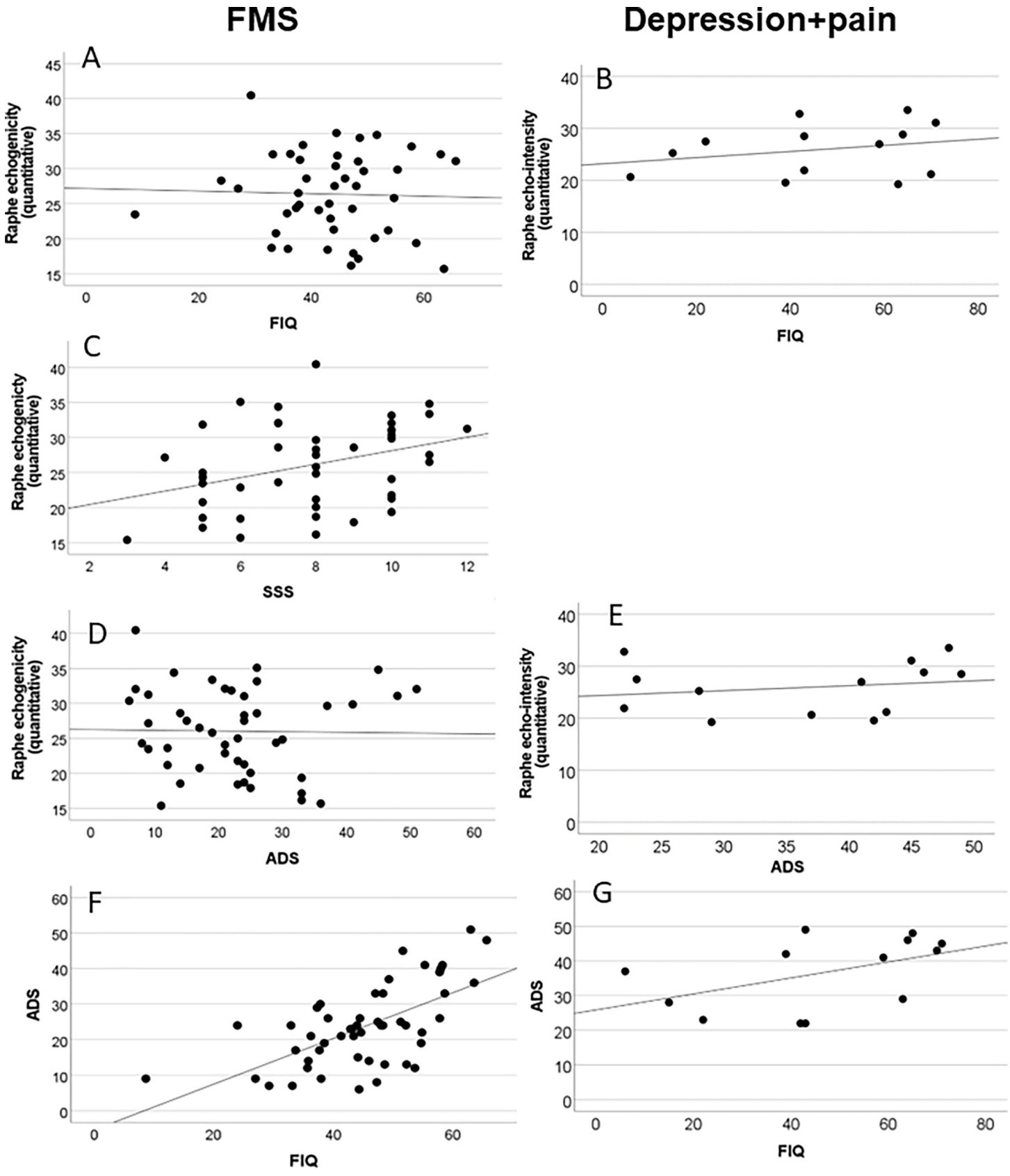

**Fig 3. Correlation analysis between midbrain raphe echo-intensity and disease burden.** The graphs show the results of the Spearman correlation analysis in patients with fibromyalgia syndrome (FMS; A, C, D, F) and depression with physical pain (B, E, G). In FMS patients (A) and in patients with depression and physical pain (B), midbrain raphe echo-intensity did not correlate with the FIQ score. In patients with FMS, midbrain raphe echo-intensity positively correlated with the FMS Symptom Severity Score (SSS, C) (correl. coeff.: 0.344; p<0.05), which was not assessed in patients with depression and physical pain. In both patient groups, the score of the "Allgemeine Depressionsskala" (ADS) did not correlate with midbrain raphe echogenicity (D, E). FIQ scores positively correlated with the ADS score in FMS patients (correl. coeff.: 0.586; p<0.001, F) and less prominently also in patients with depression and physical pain (correl. coeff.: 0.555; p = 0.049, G).

with a psychiatrically confirmed diagnosis of depression, while in other studies, mainly patients with another primary disease such as migraine [30] or Parkinson's disease [34] were studied who additionally had depressive symptoms. Hence, data may differ between patients with depression and those with mere depressive symptoms.

The major limitation of our study is that we investigated a relatively small group of patients. Hence, larger studies are needed to allow generalizable conclusions and maybe pave the way for this objective tool to enter diagnostic algorithms in clinical practice. Still, we provide evidence for a reduction of midbrain raphe echogenicity in patients with FMS compared to healthy controls. While the biological background of this finding remains to be elucidated, further studies are needed to understand the underlying pathomechanism connecting imaging data with cellular pathology.

## Supporting information

**S1 Table. Spearman correlation analysis in the group of patients with fibromyalgia syndrome.**
(DOCX)

**S2 Table. Spearman correlation analysis in the group of patients with depression and physical pain.**
(DOCX)

## Acknowledgments

We thank Johanna Frank, MD, Christoph Erbacher, PhD, Hans-Christoph Aster, MD, Franziska Gross, MD, Claudia Sommer, MD, and Daniela Urlaub (Department of Neurology, University of Würzburg, Germany) for help during subject recruitment.

## Author Contributions

**Conceptualization:** Nurcan Üçeyler, Stefan Unterecker, Uwe Walter.

**Data curation:** Nurcan Üçeyler, Mira Schließer, Dimitar Evdokimov, Jakub Radziwon, Betty Feulner, Florian Rimmele, Uwe Walter.

**Formal analysis:** Jakub Radziwon, Betty Feulner, Florian Rimmele, Uwe Walter.

**Funding acquisition:** Nurcan Üçeyler.

**Investigation:** Nurcan Üçeyler, Dimitar Evdokimov, Betty Feulner, Florian Rimmele.

**Methodology:** Nurcan Üçeyler, Mira Schließer, Florian Rimmele, Uwe Walter.

**Project administration:** Nurcan Üçeyler.

**Resources:** Nurcan Üçeyler.

**Supervision:** Nurcan Üçeyler, Uwe Walter.

**Writing – original draft:** Nurcan Üçeyler, Uwe Walter.

**Writing – review & editing:** Mira Schließer, Dimitar Evdokimov, Jakub Radziwon, Betty Feulner, Stefan Unterecker, Florian Rimmele.

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
