## [Decision Letter · Decision Letter 0]

19 Sep 2022

PONE-D-22-20467Reduced midbrain raphe echogenicity in patients with fibromyalgia syndromePLOS ONE

Dear Dr. Üçeyler,

Thank you for submitting your manuscript to PLOS ONE. After careful consideration, we feel that it has merit but does not fully meet PLOS ONE’s publication criteria as it currently stands. Therefore, we invite you to submit a revised version of the manuscript that addresses the points raised during the review process.

We look forward to receiving your revised manuscript.

Kind regards,

Keisuke Suzuki, MD, PhD

Academic Editor

PLOS ONE

Journal Requirements:

"Parts of the study was supported by the Else Kröner-Fresenius-Stiftung (N.Ü.: 2014_A129). N.Ü. was funded by Deutsche Forschungsgemeinschaft (DFG; UE171/15-1)."

"Parts of the study was supported by the Else Kröner-Fresenius-Stiftung (N.Ü.: 2014_A129). N.Ü. was funded by Deutsche Forschungsgemeinschaft (DFG; UE171/15-1)."

Additional Editor Comments:

This is an interesting study, but the sample size needs to be increased for the results to be meaningful. Please revise your manuscript carefully according to the reviewers' comments.

Reviewers' comments:

Reviewer's Responses to Questions

**Comments to the Author**

1. Is the manuscript technically sound, and do the data support the conclusions?

Reviewer #1: Partly

Reviewer #2: Yes

2. Has the statistical analysis been performed appropriately and rigorously? 

Reviewer #1: N/A

Reviewer #2: Yes

3. Have the authors made all data underlying the findings in their manuscript fully available?

Reviewer #1: Yes

Reviewer #2: Yes

4. Is the manuscript presented in an intelligible fashion and written in standard English?

Reviewer #1: Yes

Reviewer #2: Yes

5. Review Comments to the Author

Reviewer #1: It is a very interesting study that use a new classification( off ) of midbrain raphe echo-intensity, and also has new application of TCS for FMS.

There are still some issues need to be clarified.

1.How to diagnose FMS

2.why only females included in this study？

3.The number of cases is relatively small, so the conclusions drawn are not completely reliable, It is recommended to increase the sample size.

4.In this study, the ADS score of the depression + pain group was significantly higher than that of the FMS group, so a Spearman correlation analysis could not draw a true correlation between midbrain raphe echo-intensity and the existence of confounding factors.

5.Many studies revealed that reduce midbrain raphe echo-intensity was associated with depression in a series of disease.(33912243，30131761), why here no correlation was found between midbrain raphe echo-intensity and depression + pain group? The reason in the discussion is not sufficient, which need to be discussed in more depth.

Reviewer #2: This is an interesting research which is novel and has not been done in the past.

I have just a couple of minor comments:

1. in the abstract the conclusion and results looks similar, it is recommended to present a few sentences regarding the pathophysiology of the findings. I mean it is better to write briefly about the value of the findings which are presented in result section.

2.In the method please mention the criteria you selected patients in FMS group.

3.Also explain in the text about the pain group. How different they were from FMS group? Which pain they were suffering from?

4.It is better that the text be revised regarding the writing method, there are long sentences which make reading difficult.

6. PLOS authors have the option to publish the peer review history of their article (what does this mean?). If published, this will include your full peer review and any attached files.

Reviewer #1: No

Reviewer #2: No

---

## [Author Response · Author response to Decision Letter 0]

14 Oct 2022

Done.

"Parts of the study was supported by the Else Kröner-Fresenius-Stiftung (N.Ü.: 2014_A129). N.Ü. was funded by Deutsche Forschungsgemeinschaft (DFG; UE171/15-1)."

Done.

"Parts of the study was supported by the Else Kröner-Fresenius-Stiftung (N.Ü.: 2014_A129). N.Ü. was funded by Deutsche Forschungsgemeinschaft (DFG; UE171/15-1)."

Done.

Done.

Done.

Additional Editor Comments:

This is an interesting study, but the sample size needs to be increased for the results to be meaningful. Please revise your manuscript carefully according to the reviewers' comments.

We have followed the Editor`s suggestion and have added data of further 23 subjects. Respective modifications were made in the Methods and Results sections of our revised manuscript. 

Reviewers' comments:

Reviewer's Responses to Questions

Comments to the Author

1. Is the manuscript technically sound, and do the data support the conclusions?

Reviewer #1: Partly

Reviewer #2: Yes

2. Has the statistical analysis been performed appropriately and rigorously? 

Reviewer #1: N/A

Reviewer #2: Yes

3. Have the authors made all data underlying the findings in their manuscript fully available?

Reviewer #1: Yes

Reviewer #2: Yes

4. Is the manuscript presented in an intelligible fashion and written in standard English?

Reviewer #1: Yes

Reviewer #2: Yes

5. Review Comments to the Author

Reviewer #1: It is a very interesting study that use a new classification( off ) of midbrain raphe echo-intensity, and also has new application of TCS for FMS.

There are still some issues need to be clarified.

1. How to diagnose FMS

The diagnosis of FMS was made according to the Criteria of the American College of Rheumatology 1990, 2010, and the German S3 evidence-based FMS guidelines.1-3 This information was already given in the first paragraph of the Methods section of our manuscript. 

2. why only females included in this study?

We have revised our manuscript and have now also included men. Respective information are given in the Methods and Results section of our revised manuscript. 

3.The number of cases is relatively small, so the conclusions drawn are not completely reliable, It is recommended to increase the sample size.

We have followed the Reviewer´s suggestion and have increased our study cohort by data of further 23 subjects. Respective modifications were made in the Methods and Results sections of our manuscript. In addition, we performed a second reading of all midbrain sonograms by an independent reader, which yielded a high interrater reliability of raphe echogenicity assessments (Cohen’s kappa=0.85). We included this information in the Methods and Results sections of our revised manuscript. 

4.In this study, the ADS score of the depression + pain group was significantly higher than that of the FMS group, so a Spearman correlation analysis could not draw a true correlation between midbrain raphe echo-intensity and the existence of confounding factors.

ADS scores were >16 in both groups indicating clinical significance.4 A higher score in the depression group is plausible since here, depression is the main disease while in FMS, depressive symptoms accompany pain. We had pointed out this aspect in the Discussion section of our original manuscript:

“Interestingly, sonography results did not correlate with the presence and severity of depressive symptoms as quantified using the ADS sum score which was higher in patients with depression than in FMS patients (Table 1).” 

Hence, we agree that cross-correlations would not lead to reliable data and have refrained from calculating these. Data on correlation analysis presented in our manuscript merely remain within each study group. We have re-worded the respective passage in the Discussion section of our revised manuscript (please see page 8): 

“Interestingly, sonography results did not correlate with the presence and severity of depressive symptoms in each study group as quantified using the ADS sum score which was higher in patients with depression than in FMS patients (Table 1). Hence, cross-correlations are not possible.”

5.Many studies revealed that reduce midbrain raphe echo-intensity was associated with depression in a series of disease.(33912243，30131761), why here no correlation was found between midbrain raphe echo-intensity and depression + pain group? The reason in the discussion is not sufficient, which need to be discussed in more depth.

Thank you for this comment. 

33912243: This study primarily investigated patients with migraine and additional depressive symptoms using the HAM-D questionnaire. No psychiatric interview and clinical investigation was performed; hence, the diagnosis of depression cannot be made merely using one questionnaire. In contrast, we investigated patients with a confirmed psychiatric diagnosis of depression. 

30131761: This study primarily examined patients with Parkinson`s disease or essential tremor. No psychiatric diagnosis of depression was made. Merely data of the BDI questionnaire were used to screen for depressive symptoms and none of the investigated patients received anti-depressant treatment suggesting low intensity of symptoms, if any. 

We believe that great caution is necessary when cross-comparing studies reporting data of patients with a psychiatrically confirmed depression with those of patients suffering from a primarily other disease but showing hints for depressive symptoms merely assessed with questionnaires. We have added a respective passage in the Discussion section of our revised manuscript (please see page 9):

“As for the seemingly contradictory data in our study to published literature regarding reduced midbrain echogenicity in patients with depressive symptoms it is of note that we investigated patients with a psychiatrically confirmed diagnosis of depression, while in other studies, mainly patients with another primary disease such as migraine5 or Parkinson`s disease6 were studied who additionally had depressive symptoms. Hence, data may differ between patients with depression and those with mere depressive symptoms.”

Reviewer #2: This is an interesting research which is novel and has not been done in the past.

I have just a couple of minor comments:

1. in the abstract the conclusion and results looks similar, it is recommended to present a few sentences regarding the pathophysiology of the findings. I mean it is better to write briefly about the value of the findings which are presented in result section.

We have followed the Reviewer`s suggestion and have rephrased our Abstract conclusions accordingly: 

“We found reduced echogenicity of the midbrain raphe area in patients with FMS and in patients with depression and physical pain, independent of the presence or severity of pain, FMS, and depressive symptoms. Further exploration of this sonographic finding is necessary before this objective technique may enter diagnostic algorithms in FMS and depression.”

2. In the method please mention the criteria you selected patients in FMS group.

The FMS diagnosis was made according to the Criteria of the American College of Rheumatology 1990, 2010, and the German S3 evidence-based guidelines.1-3 This information was already given in the first paragraph of the Methods section of our manuscript. 

3. Also explain in the text about the pain group. How different they were from FMS group? Which pain they were suffering from?

We have followed the Reviewer`s suggestion and have added further details on the pain phenotype of the depression + pain group. Data were implemented in the Results section of our revised manuscript. 

4. It is better that the text be revised regarding the writing method, there are long sentences which make reading difficult.

Thank you. We have carefully revised our manuscript accordingly. 

6. PLOS authors have the option to publish the peer review history of their article (what does this mean?). If published, this will include your full peer review and any attached files.

Do you want your identity to be public for this peer review? For information about this choice, including consent withdrawal, please see our Privacy Policy.

Reviewer #1: No

Reviewer #2: No

References: 

1. Eich, W. et al., Z Rheumatol 2008;67:665-666, 668-672, 674-666.

2. Wolfe, F. et al., Arthritis Care Res (Hoboken) 2010;62:600-610.

3. Wolfe, F. et al., Arthritis Rheum 1990;33:160-172.

4. Radloff, L. S., Appl Psychol Meas 1977;3:385–401.

5. Zhang, Y. et al., Ther Adv Neurol Disord 2021;14:17562864211007708.

6. Richter, D. et al., Front Neurol 2018;9:645.

---

## [Decision Letter · Decision Letter 1]

25 Oct 2022

Reduced midbrain raphe echogenicity in patients with fibromyalgia syndrome

PONE-D-22-20467R1

Dear Dr. Üçeyler,

We’re pleased to inform you that your manuscript has been judged scientifically suitable for publication and will be formally accepted for publication once it meets all outstanding technical requirements.

Kind regards,

Keisuke Suzuki, MD, PhD

Academic Editor

PLOS ONE

Additional Editor Comments (optional):

Reviewers' comments:

Reviewer's Responses to Questions

**Comments to the Author**

1. If the authors have adequately addressed your comments raised in a previous round of review and you feel that this manuscript is now acceptable for publication, you may indicate that here to bypass the “Comments to the Author” section, enter your conflict of interest statement in the “Confidential to Editor” section, and submit your "Accept" recommendation.

Reviewer #1: All comments have been addressed

2. Is the manuscript technically sound, and do the data support the conclusions?

Reviewer #1: Yes

3. Has the statistical analysis been performed appropriately and rigorously? 

Reviewer #1: Yes

4. Have the authors made all data underlying the findings in their manuscript fully available?

Reviewer #1: Yes

5. Is the manuscript presented in an intelligible fashion and written in standard English?

Reviewer #1: Yes

6. Review Comments to the Author

Reviewer #1: (No Response)

7. PLOS authors have the option to publish the peer review history of their article (what does this mean?). If published, this will include your full peer review and any attached files.

Reviewer #1: No

---

## [Editor Report · Acceptance letter]

8 Nov 2022

PONE-D-22-20467R1 

Reduced midbrain raphe echogenicity in patients with fibromyalgia syndrome 

Dear Dr. Üçeyler:

I'm pleased to inform you that your manuscript has been deemed suitable for publication in PLOS ONE. Congratulations! Your manuscript is now with our production department. 

Kind regards, 

on behalf of

Dr. Keisuke Suzuki 

Academic Editor

PLOS ONE